# Screening of the Antimelanoma Activity of Monoterpenes—In Vitro Experiments on Four Human Melanoma Lines

**DOI:** 10.3390/cimb47020097

**Published:** 2025-02-03

**Authors:** Paula Wróblewska-Łuczka, Laura Kulenty, Katarzyna Załuska-Ogryzek, Agnieszka Góralczyk, Jarogniew J. Łuszczki

**Affiliations:** Department of Occupational Medicine, Medical University of Lublin, 20-090 Lublin, Polandkatarzyna.zaluska-ogryzek@umlub.pl (K.Z.-O.); agnieszka.goralczyk@umlub.pl (A.G.); jarogniew.luszczki@umlub.pl (J.J.Ł.)

**Keywords:** melanoma, monoterpenes, citral, myrcene, citronellol, rhodinol, geraniol, nerol, thymol, carvacrol

## Abstract

(1) Malignant melanoma is the most aggressive type of malignant tumor caused by a dysfunction of melanocytes. Despite progress in the treatment of melanoma, further research and search for new potential drugs are necessary to optimize the therapy. (2) The aim of this study was to evaluate the antiproliferative activity of eight selected monoterpenes by MTT and LDH assays on four malignant melanoma cell lines. (3) Myrcene, rhodinol and nerol did not show any significant anticancer effect on melanoma cell lines, but citral, carvacrol, citronellol, thymol and geraniol showed a significant anti-viability effect. Our studies have shown that the most effective terpene among those tested in inhibiting melanoma cell viability was carvacrol, with the lowest IC50 in the range of 0.05 ± 0.00 to 0.06 ± 0.01 mM. Moreover, it did not negatively affect normal human keratinocyte cells. (4) Metastatic melanoma is very difficult to treat, and some terpenes have the ability to sensitize cells to other chemicals; so, it is worth investigating their antimelanoma potential, as terpenes could become an adjuvant to traditional treatment.

## 1. Introduction

Malignant melanoma is the most aggressive type of malignant tumor caused by a dysfunction of pigment cells (melanocytes) that are present not only in mucous membranes, eyeballs and hair follicles, but also in the skin, becoming a global problem due to the constant increase in the incidence of the disease worldwide. It occurs mainly in the white population. People with fair skin, light-colored eyes and red or blond hair are particularly at risk. The main method of melanoma treatment is surgery, but relapses or metastasis are often observed and they are difficult to treat [1,2]. Despite progress in the treatment of melanoma, further research and search for new potential drugs are necessary to optimize the therapy.

Terpenes are naturally occurring substances produced by many plants. These hydrocarbons are also one of the main components of resins and essential oils and are often responsible for the specific smell of the plant, acting as attractants or repellents. The physiological function of these secondary metabolites is to participate in the plant’s resistance against pathogens [3].

A huge number of structures and functions of terpenes and their derivatives arouse great scientific interest. Some of these compounds have found commercial application in the prevention and therapy of many diseases due to their antibacterial, antifungal, antiparasitic, antiviral, antiallergic, anticonvulsant, anti-inflammatory and immunomodulatory properties [4]. Some terpenes are used in anticancer therapy as registered drugs, e.g., paclitaxel from the taxane group [5].

All terpenes selected for testing herein are characterized by a number of biological activities, including anticancer properties. For example, myrcene is characterized by anti-inflammatory, analgesic and antioxidant activity [6]. Due to its antioxidant nature, it can also counteract aging processes. This is evidenced by a study on human fibroblasts, which showed that by regulating the synthesis of extracellular matrix metalloproteinases and inhibiting the MAPK pathway, myrcene slows down the skin photoaging process [7]. Additionally, myrcene inhibits the growth of cervical cancer, lung cancer [8] and liver cancer [9]. Citronellol inhibits the proliferation of breast cancer [10,11,12] and non-small-cell lung cancer [13,14]. Nerol and geraniol also exhibit anticancer properties [15], although geraniol is much more effective and inhibits the growth of colon cancer [16], liver cancer [17], endometrial cancer [18], breast cancer [19], and prostate cancer [20]. Citral has anticancer activity against breast cancer, lung cancer, and colon cancer [21,22,23,24,25]. Thymol and its isomer carvacrol also exhibit antiproliferative effects. Thymol is active against non-small-cell lung cancer [26], breast cancer and leukemia cells [27], and mouse and human melanoma [28,29]. Carvacrol, like thymol, also has anticancer activity against breast cancer [30], non-small-cell lung cancer [31], prostate cancer and cervical cancer [32,33,34]. Generally, the mechanism of action of monoterpenes in anticancer activity is associated with the induction of apoptosis, increased production of ROS and arrest of the cell cycle [24,25,26,32,33,34,35,36]. Differences in the chemical structure of the compounds are shown in Figure 1.

The aim of this study was to evaluate the antiproliferative activity of selected monoterpenes, i.e., myrcene (acyclic monoterpene), citral (terpene aldehyde), thymol and its isomer carvacrol (phenolic monoterpenes), geraniol and its cis isomer nerol, citronellol and its left-side isoform rhodinol (monoterpene alcohols) against four human melanoma cell lines. When selecting terpenes for research, the search was narrowed to monoterpenes derived from citrus plant essential oils, trying to select those better known in nature and present in larger quantities in essential oil (as well as their isomers, if any). This publication presents a new perspective on the search for monoterpenes active against melanoma, comparing the results assessing the effect of compounds on the metabolic activity of cells (MTT test) as well as the assessment of the cytotoxicity of compounds both against melanoma and the normal keratinocyte cell line. The experiments also included lesser-known terpenes (being isomers of better-known ones) for which there are no data on their anticancer properties, i.e., myrcene, nerol and rhodinol.

## 2. Materials and Methods

### 2.1. Malignant Melanoma Cell Culture

Four malignant melanoma cell lines, FM55P, FM55M2 (ECACC, Public Health England, Porton Down, UK), A375 and SK-MEL28 (ATCC, Manassas, VA, USA) were used. The immortalized human normal keratinocyte (HaCaT; ATCC, Manassas, VA, USA) served as a reference cell line. Of note, the A375 cell line is the original and not a CRISPR-modified version of the cell line, and all the tested cell lines in this study have the BRAF V600E mutation. The experimental cell culture conditions were as follows: 37 °C in a humidified atmosphere of 95% air and 5% CO_2_. The culture media used were as follows: for A375—DMEM high glucose; for SK-MEL28—EMEM; and for FM55P and FM55M2—RPMI1640 (All from Sigma-Aldrich, St. Louis, MO, USA). Previous publications described in more detail the growth conditions of these cell line cultures [37,38].

### 2.2. Tested Drugs

Citral, myrcene, thymol and carvacrol were dissolved in DMSO. Citronellol, rhodinol, geraniol, and nerol were dissolved in EtOH (all the drugs and solvents were from Sigma-Aldrich, St. Louis, MO, USA). The compounds were dissolved to the tested concentrations in a growth medium before being added to the microtiter (96-well) plate. DMSO and EtOH were maintained at the safe concentration of 0.1% and had no effect on cell growth.

### 2.3. Cell Viability Assessment—MTT Test

The effect on cell viability of the tested terpenes, citral, myrcene, citronellol, rhodinol, geraniol, nerol, thymol and carvacrol, was assessed using the MTT assay. All tested cell lines—the HaCaT cell line model of human keratinocyte cells [39] (density: 1 × 10^4^ cells/mL) and four human malignant melanoma lines (density: 2–3 × 10^4^ cells/mL, depending on the cell line—were plated on microtiter plates (NEST Biotechnology, Wuxi, China). After a day (24 h) of incubation, the medium was replaced with a fresh medium to which increasing concentrations of the tested substances were added. The experimental steps of the MTT assay were previously described [37]. Each experiment in the MTT assay was performed in triplicate to ensure the repeatability and validity of the results.

### 2.4. Cell Cytotoxicity—LDH Test

The cytotoxicity of the tested terpenes that had a significant effect on the proliferation of melanoma lines in the MTT test—citral, citronellol, geraniol, thymol and carvacrol—was assessed using the LDH assay (Cytotoxicity Detection KitPLUS LDH, Roche Diagnostics, Mannheim, Germany) based on the measurement of lactate dehydrogenase activity released into the medium from damaged cells. Specified densities (as above) of all the studied melanoma and normal human cell line were plated on microtiter plates (NEST). After 24 h, the medium was replaced with a fresh medium supplemented with the tested terpenes: citral, citronellol, geraniol, thymol and carvacrol (in the same concentrations for individual substances as in the MTT test). The LDH test was carried out after 72 h of incubation with the tested terpenes according to the manufacturer’s instructions. A more detailed description of the steps of the LDH test was previously provided [37].

### 2.5. Statistical Analysis

GraphPad Prism (version 8.0, San Diego, CA, USA) was used for statistical analysis of the data (from the MTT and LDH tests). The analyses were performed with a one-way ANOVA test followed by Tukey’s post hoc test. Data were presented as means ± standard errors (SEMs). The median inhibitory concentration (IC_50_) values for the tested terpenes were calculated automatically with a computer-assisted log-probit method using an MS Excel spreadsheet, as previously described [38,40].

## 3. Results

### 3.1. The MTT Assay

In the MTT test, it was observed that five out of the eight tested terpenes, i.e., citral, carvacrol, citronellol, thymol and geraniol, showed a significant anti-viability effect on melanoma cells. It was possible to determine the IC_50_ for these compounds in the range of the tested concentrations that could be obtained taking into account the solubility of the compound. The tested terpenes inhibited the viability of four malignant melanoma cell lines (Figure 2a–e). Additionally, the effect of the tested terpenes on the viability of human normal keratinocytes was reported (Figure 2a–e).

The terpenes citral, carvacrol, citronellol, thymol and geraniol showed a significant effect on the viability of four human melanoma lines in the range of almost all tested concentrations of the individual compounds (Figure 2a–e). It is worth noting that citronellol was found to be the least safe for human keratinocytes, as it, in the range of all tested concentrations (1.2–6.0 mM), significantly affected their proliferation (Figure 2c). Citral was found to be equally toxic for HaCaT cells, inhibiting their proliferation by up to 28% at a concentration of 0.3 mM and by up to approximately 10% at the highest concentration of 0.4 mM (Figure 2a). Carvacrol was found to be the safest for normal cells, as it did not significantly affect cell viability up to a concentration of 0.65 mM (Figure 2b).

The experimentally derived median inhibitory concentration (IC_50_) values for the tested terpenes in various melanoma cell lines are presented in Table 1.

To experimentally assess the cytotoxic effects of the tested terpenes on normal human cells and to determine their safety profile, the selectivity index, as a ratio of the IC50 for normal cell lines (HaCaT) and IC50 for the respective melanoma cell lines (A375, SK-MEL 28, FM55P and FM55M2), was calculated. Generally, the tested compounds with selectivity indices higher than 1 indicate drugs with greater efficacy against tumor cells than toxicity against normal cells [41]. In this study, the selectivity index for the tested terpenes was higher than 1 concerning keratinocytes (HaCaT cell line). Carvacrol was found to be the safest for normal lines, with a selectivity index above 10 (Figure 3).

In contrast, myrcene, rhodinol and nerol did not show any significant anticancer effect on melanoma lines (Figure 4a–c). Myrcene in the range of the tested concentrations up to 5mM did not show any significant inhibition of melanoma line growth, and interestingly, in concentrations of 3 mM, it had a statistically significant effect on the viability of normal keratinocyte lines (Figure 4a). Rhodinol inhibited the viability of both melanoma cells and normal keratinocytes from a dose of 4.8 mM (Figure 4b). Nerol significantly affected the viability of the tested melanoma cell lines in the range of all tested concentrations (0.13–0.65 mM). The effect of nerol on normal keratinocyte HaCaT cells was noted only at the highest concentration of 0.65 mM (Figure 4c). For these three terpenes (myrcene, rhodinol, nerol), the IC_50_ values could not be determined taking into account their solubility and the safe dose of DMSO/EtOH; therefore, LDH tests for these compounds were not performed.

### 3.2. The LDH Assay

The cytotoxicity of the five tested terpenes, citral, carvacrol, citronellol, thymol and geraniol, to normal human keratinocytes (HaCaT) and four malignant melanoma cell lines (A375, SK-MEL28, FM55P, FM55M2) was detected by means of the LDH assay.

All tested terpenes showed statistically significant cytotoxicity towards melanoma cells: citral at concentrations above 0.2 mM (Figure 5a), carvacrol in the range of 0.05–0.65 mM depending on the cell line (Figure 5b), citronellol at concentrations above 4.2 mM (Figure 5c), thymol at concentrations above 0.03 mM (Figure 5d), and geraniol at concentrations above 0.32 mM, but only in the case of two cell lines, FM55P and FM55M2 (Figure 5e). Thymol (Figure 5d) seems to be the most cytotoxic of the tested terpenes for melanoma cells as its statistically significant activity was observed for all cell lines at most of the tested concentrations (in the range of 0.05 to 1.30 mM, and in the case of the SK-MEL28 line from a concentration of 0.03 mM). It is worth noting that none of the tested terpenes showed cytotoxicity towards the normal human keratinocyte HaCaT line (Figure 5a–e).

## 4. Discussion

Our experiments have shown that myrcene had the worst antimelanoma properties among the tested terpenes. Studies on rats have also failed to confirm that myrcene administered in the diet inhibits carcinogenesis [42]. The cell lines on which the effect of myrcene was verified were human cervical cancer, human lung cancer, monkey kidney cells and mouse macrophages. However, myrcene at a concentration of 200 µg/mL (≈1.5 mM) had no cytotoxic activity against any cell line [8]. In vitro experiments have shown that myrcene inhibited cell proliferation (IC_50_ in the range of 9.23 to 12.27 μg/mL (≈0.07–0.09 mM) for liver cancer cell lines and mouse melanoma B16-F10 [9], which, unfortunately, was not confirmed in our studies. Interestingly, essential oils of plants in their full composition are characterized by significantly higher anticancer activity than myrcene itself, from which it was isolated [8,9].

Another terpene, rhodinol, did not show any significant activity against human melanoma cells. There are no reports in the literature of its possible anticancer activity, but we wanted to verify whether rhodinol (sometimes called α-citronellol) has similar effects to those of its isomer, β-citronellol.

Our experiments have shown that citronellol significantly inhibits melanoma proliferation, but among the terpenes we tested, the highest IC_50_ value was for citronellol (range 2.17 ± 0.30 mM to 4.81 ± 0.29 mM depending on the cell line). Scientists have demonstrated anticancer activity in breast cancer in vitro and in vivo models, finding that citronellol reduced inflammation, inhibited cancer cell evolution, reduced and eliminated tumors, and improved the general condition of experimental animals [10,11,12]. In experiments on the non-small-cell lung cancer cell line, citronellol was shown to promote tumor cell death, and other experiments showed that it prevents infections in patients during and after systemic therapy [13,14]. Evaluation of the effect of citronellol in triple-negative breast cancer showed cytotoxicity, and that it reduced proliferation, inhibited migration, reduced colony formation, arrested cell cycle in the G2/M phase and initiated apoptosis [11]. Similar effects of the substance were found in another study of human breast cancer lines. The study determined the IC_50_ for the MCF-7 line at 0.08 mM and 0.035 mM for MDA-MB-231 cells [10]. The results of our studies indicate that the IC_50_ values of citronellol against melanoma are much higher (2.17 ± 0.30 mM to 4.81 ± 0.29 mM) compared to those obtained for breast cancer cells. An interesting finding is the consequence of using citronellol as an adjuvant in chemo- and radiotherapy [14].

Nerol tested in our experiments showed activity against human melanoma cells. Its effect was not significant, and it was not possible to determine the IC_50_ value of the compound in the range of the tested concentrations. There are some reports confirming its anticancer effect [15], but its trans isomer, geraniol, has been studied much more often.

Geraniol showed an antitumor effect in the colo-205 colon cancer cell line (IC_50_ = 0.02 mM). The mechanism of action was due to the induction of apoptosis, causing damage in the tumor DNA and leading to the arrest of the cell cycle in the G2/M phase [16]. Other data showed that geraniol inhibited the proliferation of liver cancer cells lines [17]. Studies performed using the Ishikawa cell line, of endometrial cancer, showed that the cell population decreased depending on the geraniol concentrations administered. The IC_50_ was 0.26 mM for 24 h of incubation. Geraniol led to a decrease in Bcl-2 expression, with a simultaneous increase in the genes encoding Bax protein and caspases 3 and 8, thus leading to the intensification of the apoptosis process in cells [18]. In turn, studies of changes in the MCF-7 breast cancer cell population showed that geraniol inhibits their growth depending on the concentration (range 0.1–0.7 mM) and time (2 to 10 days) [19]. Scientists have demonstrated the antitumor activity of geraniol against mouse melanoma B16(F10) cells with an IC_50_ of 150 ± 19 mM, additionally demonstrating in vivo that the addition of natural compounds (including geraniol) to the mouse diet inhibits the growth of neoplastic lesions [43,44,45]. The IC50 values obtained herein are slightly higher (IC_50_ in the range of 0.27 ± 0.03 mM to 0.44 ± 0.03 mM), which may be due to differences in sensitivity between mouse and human cells. In vitro tests conducted on the A431 skin cancer line showed an IC_50_ of 0.098 mM for geraniol in the MTT assay. Cell cycle analyses confirmed that geraniol arrested the G0/G1 phase of A431 cells in a dose-dependent manner. In addition, geraniol was found to inhibit the growth of Ehrlich Ascites Carcinoma (EAC) in Swiss albino mice and to be safe up to 1000 mg/kg body weight in an acute oral toxicity study [46].

Our experiments indicated that citral significantly affected the viability of human melanoma cells, and the IC50 values we obtained ranged from 0.08 ± 0.00 mM to 0.13 ± 0.01 mM depending on the cell line. Unfortunately, at high concentrations, it had a negative effect on normal keratinocyte cells. Studies conducted on lung cancer, breast cancer and colon adenocarcinoma cell lines emphasize the potential anticancer activity of lemongrass oil, an important component of which is citral. The studies also showed that high concentrations of the compound inhibit the migration and proliferation of cancer cells [21,22,23,24,25]. The effect of citral on reducing cell viability has already been studied in breast cancer. The IC_50_ values for breast cancer cell lines were 20.5 ± 1.41 µg/mL (≈0.135 mM) and 21.4 ± 2.82 µg/mL (≈0.14 mM) in MDA MB-231 and 4T1 cell lines, respectively [24,47], which are comparable to those obtained in our work. In comparison, for the A549 and H1299 lung cancer cell lines, inhibition of cell proliferation occurred for citral with IC_50_ values of 1.73 ± 0.37 μg/mL (≈0.011 mM) and 4.01 ± 0.30 μg/mL (≈0.026 mM), respectively [25], indicating a much higher sensitivity of this cancer to citral. The IC_50_ values determined for colon cancer cell lines were 22.50 ± 2.50 µg/mL (≈0.148 mM) for the SW620 line and 21.77 ± 0.23 µg/mL (≈0.143 mM) for the HT29 line [48]. Numerous studies conducted on cancer cell lines allowed for the observation of apoptosis induction in cancer cells by citral [24,25,35,36].

Thymol has a proapoptotic effect, causing cell damage and mitochondrial dysfunction [49,50]. Thymol led to cell cycle arrest and initiation of apoptosis in non-small-cell lung cancer cells and oral squamous cell carcinoma cells. Moreover, researchers confirmed its lack of a cytotoxic effect on healthy cells [26,51]. In the case of breast cancer and leukemia cells, it was confirmed that thymol leads to cell cycle arrest in the G0/G1 phase. In the case of glioblastoma multiforme or laryngeal cancer, it leads to cell death by apoptosis or necrosis [27]. Thymol has also been studied as a protective compound against harmful UVA and UVB solar radiation. This is of particular importance in relation to melanoma, because its risk factor is related to exposure to excessive solar radiation. In a study conducted on healthy keratinocytes, thymol was shown to inhibit the production of reactive oxygen species (ROS) in cells irradiated with both UVA and UVB, and reduced DNA damage [52,53]. The anticancer properties of thymol were also studied on the B16-F10 mouse melanoma cell lines. The assessment of cell viability using the MTT test showed that the IC_50_ value = 60.09 µg/mL (0.4 mM) [29], which means that this value is lower than the IC_50_ values for thymol that we obtained in the study, in the range of 0.46 ± 0.05 mM to 0.53 ± 0.05 mM depending on the human melanoma cell line. In another study, it was found that essential oil from Thymus vulgaris (in which the composition of thymol comprised 24.7%) at a concentration of 100 µg/mL inhibited the growth of 30% of the mouse melanoma cell line B164A5 and 27% of the human melanoma cell line A375 [28]. These results are difficult to compare with our own results due to the use of the whole essential oil and not the pure substance (thymol). Our study confirms that thymol has an inhibitory effect on the viability of human melanoma cells; however, at high concentrations, it affects the viability of normal cell lines without cytotoxicity.

Our studies have shown that the most effective terpene among those tested in inhibiting melanoma cell viability was carvacrol, with the lowest IC_50_ in the range of 0.05 ± 0.00 to 0.06 ± 0.01 mM. All tested melanoma cell lines responded to different concentrations of the compound in a very comparable way. Moreover, it did not negatively affect normal cells. The anticancer properties of carvacrol have also been demonstrated in relation to breast cancer. This compound is able to inhibit cell progression in a dose- and time-dependent manner. Inhibition of cell cycle progression (G0/G1 phase) has also been proven by reducing the expression of certain cycle proteins [54]. In a study on non-small-cell lung cancer (lines A549 and H460), an inhibitory effect of carvacrol on the proliferation and migration of cancer cells was indicated [31]. In studies on lung adenocarcinoma, carvacrol was shown to inhibit proliferation and reduce the migration of cancer cells, IC_50_ = 100 μg/mL (≈0.67 mM) for the A549 line [55]. Scientists have examined the effect of essential oil from *Nepeta glomerata*, in the composition of which significant amounts of carvacrol were identified. This oil, at a concentration of 100 mg/mL, inhibited the proliferation of C32 cells—colorless human melanoma—by 48% [56]. The results obtained by the authors are difficult to compare with the results obtained in this study due to the use of essential oil and not pure carvacrol. Another experiment showed the antiproliferative effect of carvacrol on the murine melanoma cell line B16F10; the IC_50_ of carvacrol was 120 ± 15 µmol/L (120 mM) [43], which is a significantly higher result compared to the results of this study, which may be related to the greater sensitivity of human melanoma cell lines to carvacrol. Interestingly, other researchers verified the effect of carvacrol used in nanoemulsion on the proliferation of A375 melanoma cells, and the IC_50_ = 100.24 µg/mL (which is approximately 0.66 mM) obtained by them is about 10 times higher than that obtained in this study, which may be due to the use of various additives in the nanoemulsion, such as medium-chain triglyceride (MCT), lecithin and non-ionic surfactant and polysorbate 80 emulsifier, which could affect proliferation [57].

## 5. Conclusions

Terpenes are the subject of biochemical and molecular studies due to their numerous biological activities, including anti-inflammatory, antimicrobial and antiviral effects. The anticancer properties of terpenes have also been widely studied in recent years. Numerous studies indicate their anticancer potential against many types of cancers, including melanoma. Terpenes interact with specific biological targets, such as receptors, ion channels, and enzymes, and can also modulate signaling pathways involved in various cellular processes (apoptosis, cell proliferation and differentiation). Metastatic melanoma is very difficult to treat, and some terpenes have the ability to sensitize cells to other chemicals; so, it is worth investigating their antimelanoma potential as terpenes could become an adjuvant to traditional treatment.

## Figures and Tables

**Figure 1 cimb-47-00097-f001:**
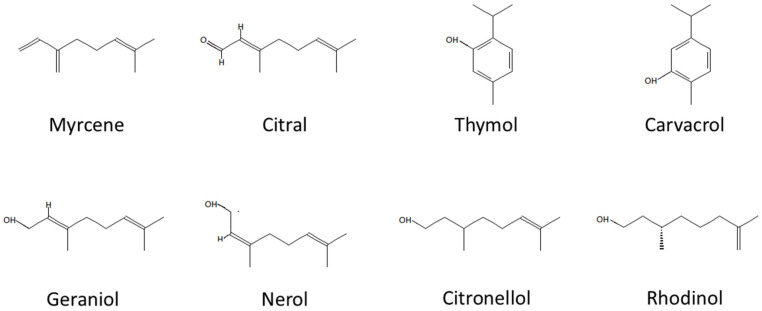
Chemical structure of the monoterpenes tested.

**Figure 2 cimb-47-00097-f002:**
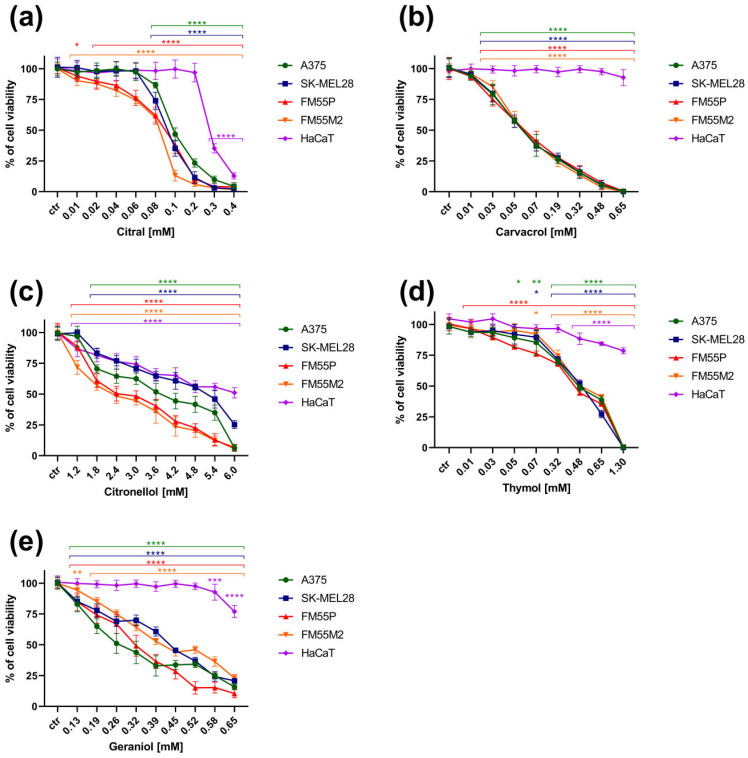
Influence of increasing concentrations of the tested terpenes (citral (**a**), carvacrol (**b**), citronellol (**c**), thymol (**d**), geraniol (**e**)) on the cell viability of malignant melanoma cell lines A375, SK-MEL28, FM55P, FM55M2 and normal human keratinocytes HaCaT in the MTT assay after 72 h. Plots represent means ± SEM (as error bars). * *p* < 0.05; ** *p* < 0.01; *** *p* < 0.001 and **** *p* < 0.0001 vs. the control (ctr) group. n = 8 (for one experiment; each experiment was repeated three times).

**Figure 3 cimb-47-00097-f003:**
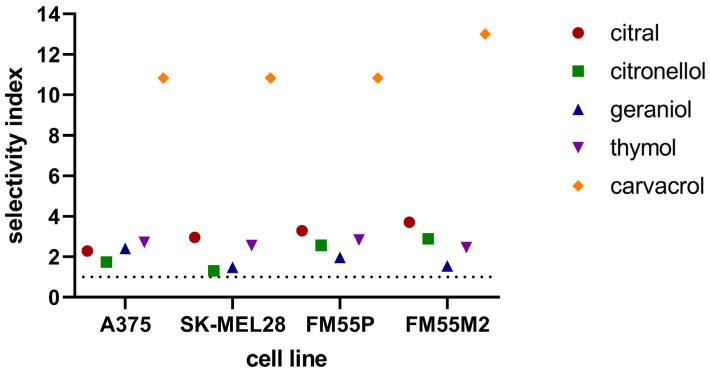
Comparison of selectivity index for AM1172 in the MTT assay. The selectivity index was calculated using the following formula: SI = (IC_50_ for normal cell line)/(IC_50_ for respective melanoma cell line).

**Figure 4 cimb-47-00097-f004:**
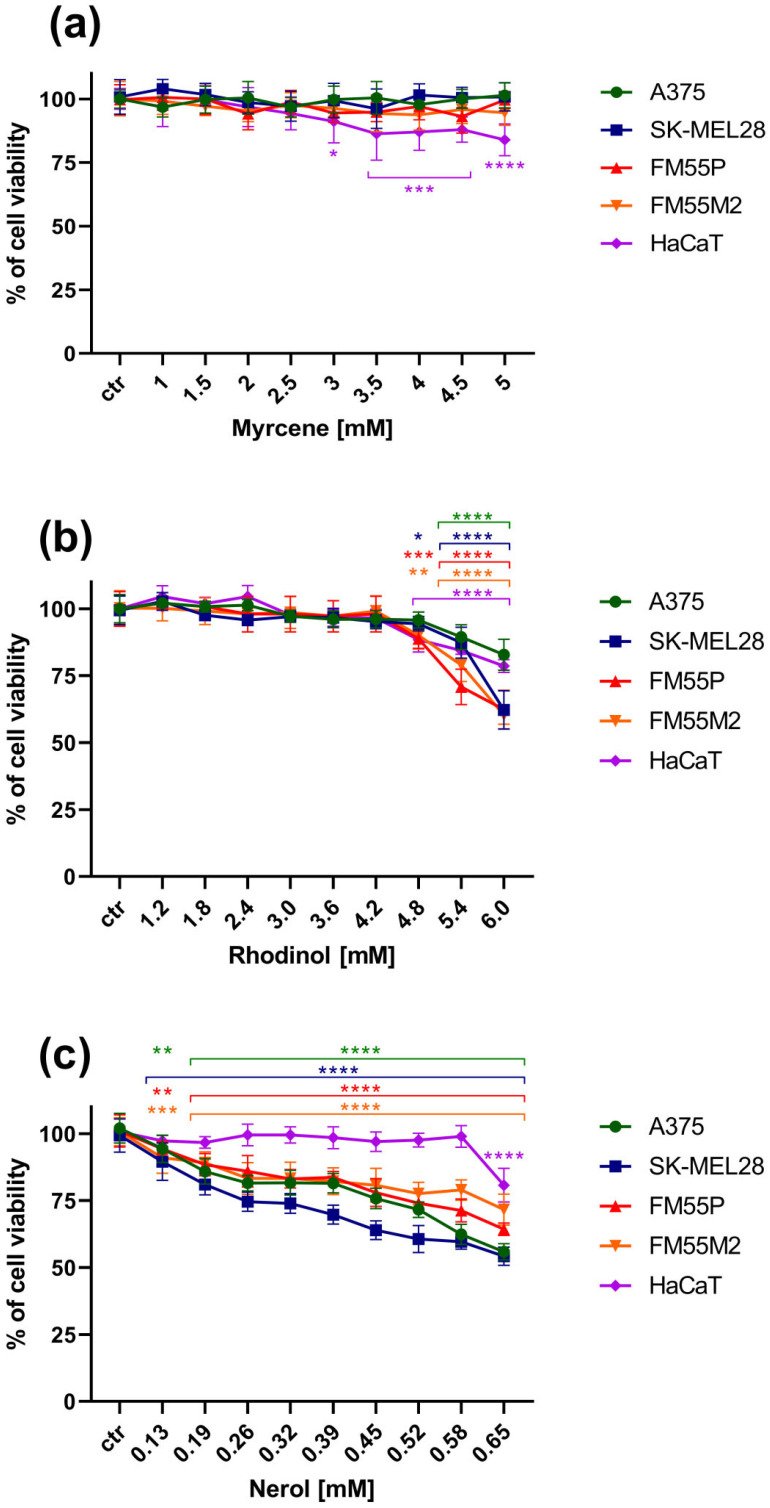
Influence of increasing concentrations of the tested terpenes (myrcene (**a**), rhodinol (**b**), nerol (**c**)) on the cell viability of malignant melanoma cell lines A375, SK-MEL28, FM55P, FM55M2 in the MTT assay after 72 h. Plots represent means ± SEM (as error bars). * *p* < 0.05, ** *p* < 0.01, *** *p* < 0.001 and **** *p* < 0.0001 vs. the control (ctr) group. n = 8 (for one experiment; each experiment was repeated three times).

**Figure 5 cimb-47-00097-f005:**
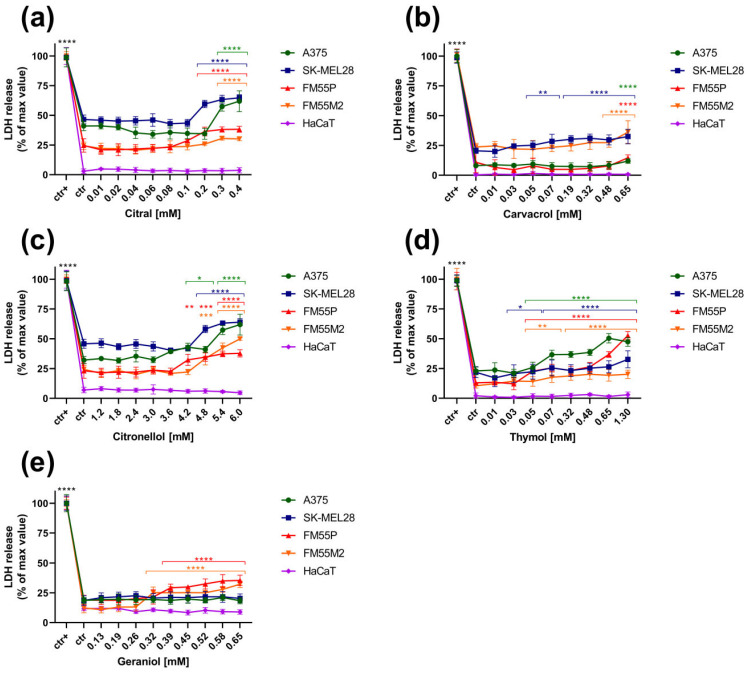
Influence of increasing concentrations of the tested terpenes (citral (**a**), carvacrol (**b**), citronellol (**c**), thymol (**d**), geraniol (**e**)) on cytotoxicity in malignant melanoma cell lines: A375, SK-MEL28, FM55P, FM55M2, normal human keratinocytes HaCaT in the LDH assay. Plots represent means ± SEM (as error bars). * *p* < 0.05, ** *p* < 0.01, *** *p* < 0.001 and **** *p* < 0.0001 vs. the control (ctr) group; ctr+—cells treated with lysis buffer. n = 8 (for one experiment; each experiment was repeated three times).

**Table 1 cimb-47-00097-t001:** The antiproliferative effects of tested terpenes in four various malignant melanoma cell lines detected in vitro by the MTT assay. The IC_50_ values are presented as means ± SEM.

Cell Line/ Terpenes [mM]	A375	SK-MEL28	FM55P	FM55M2	HaCaT
Citral	0.13 ± 0.01	0.10 ± 0.01	0.09 ± 0.01	0.08 ± 0.00	0.29 ± 0.01
Citronellol	3.63 ± 0.56	4.81 ± 0.29	2.45 ± 0.38	2.17 ± 0.30	6.27 ± 1.15
Geraniol	0.27 ± 0.03	0.44 ± 0.03	0.33 ± 0.02	0.42 ± 0.02	>0.65
Thymol	0.48 ± 0.04	0.51 ± 0.03	0.46 ± 0.05	0.53 ± 0.05	>1.3
Carvacrol	0.06 ± 0.00	0.06 ± 0.01	0.06 ± 0.01	0.05 ± 0.00	>0.65

## Data Availability

The raw data supporting the conclusions of this article will be made available by the authors on request.

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
