# Peer review of "Screening of the Antimelanoma Activity of Monoterpenes—In Vitro Experiments on Four Human Melanoma Lines"

_cimb, 2025, doi:10.3390/cimb47020097_

Round 1
Reviewer 1 Report
Comments and Suggestions for Authors
MAJOR COMMENTS
1. The rationale behind the choice of the terpenes to be tested is not clear. Why these particular terpenes?
2. There are several studies in literature that evaluate the antimelanoma activity in vitro of most of the terpenes reported here. What’s the novelty? The authors should provide an overview of the state of the art and highlight the new contribution of their work.
MINOR COMMENTS
1. I suggest to add a figure with the chemical structures of the syudied terpenes.
2. In the discussion section, I suggest to express the IC50 values all in mM, instead of ug/mL
Author Response
MAJOR COMMENTS
- The rationale behind the choice of the terpenes to be tested is not clear. Why these particular terpenes?
Response 1: An extensive explanation has been added in the introduction.
„When selecting terpenes for research, the search was narrowed to monoterpenes derived from citrus plant essential oils, trying to select those better known in nature and present in larger quantities in essential oil (plus their isomers, if any).”
- There are several studies in literature that evaluate the antimelanoma activity in vitro of most of the terpenes reported here. What’s the novelty? The authors should provide an overview of the state of the art and highlight the new contribution of their work.
Response 2: An extensive explanation has been added in the introduction.
„The publication presents a new perspective on the search for monoterpenes active against melanoma, comparing the results assessing the effect of compounds on the metabolic activity of cells (MTT test) as well as the assessment of the cytotoxicity of compounds both againsth melanoma and the normal keratinocyte cell line. The experiments also included lesser-known terpenes (being isomers of better known ones) for which there is no data on their anticancer properties, i.e. myrcene, nerol or rhodinol.”
MINOR COMMENTS
- I suggest to add a figure with the chemical structures of the studied terpenes.
Response: Thank you for this comment. The authors have added a figure in the introduction showing the chemical structure of individual monoterpenes, as suggested.
- In the discussion section, I suggest to express the IC50 values all in mM, instead of ug/mL
Response: The authors in the discussion section included results obtained by other authors and presented them in the same unit as in the source work. The conversion factor was included and the values were given in the mM unit, as suggested.
Thank You very much for Your time for the Review and all critical comments. We hope that our responses to the objections are sufficient and satisfactory for the Reviewer.
Reviewer 2 Report
Comments and Suggestions for Authors
This interesting study highlights the antiproliferative effects of various monoterpenes on melanoma cell lines, with carvacrol showing the most significant activity, evidenced by the lowest IC50 values.The manuscript is well written. Also, the results were clearly presented.
Minor changes are proposed:
Could the authors explain more analytically in the aim of the study why they selected the specific terpenes?
In the caption of Figure 1, it is suggested to add the number of replicates in each cell line. The same stands for Figure 3 and 4
Have the authors studied combinations of the terpenes in order to improve the efficacy against melanoma?
Author Response
This interesting study highlights the antiproliferative effects of various monoterpenes on melanoma cell lines, with carvacrol showing the most significant activity, evidenced by the lowest IC50 values.The manuscript is well written. Also, the results were clearly presented.
Minor changes are proposed:
Could the authors explain more analytically in the aim of the study why they selected the specific terpenes?
Response: An extensive explanation has been added in the introduction.
„When selecting terpenes for research, the search was narrowed to monoterpenes derived from citrus plant essential oils, trying to select those better known in nature and present in larger quantities in essential oil (plus their isomers, if any). The publication presents a new perspective on the search for monoterpenes active against melanoma, comparing the results assessing the effect of compounds on the metabolic activity of cells (MTT test) as well as the assessment of the cytotoxicity of compounds both againsth melanoma and the normal keratinocyte cell line. The experiments also included lesser-known terpenes (being isomers of better known ones) for which there is no data on their anticancer properties, i.e. myrcene, nerol or rhodinol.”
In the caption of Figure 1, it is suggested to add the number of replicates in each cell line. The same stands for Figure 3 and 4
Response: Thank you for this comment. The information about the number n has been added in the figure captions. We clarify that n=8 (for one experiment), but each experiment was repeated three times. So the data on the graph are the average of 24 measurements performed for a single concentration of the tested compound.
Have the authors studied combinations of the terpenes in order to improve the efficacy against melanoma?
Response: Thank you for this comment. The presented publication was supposed to present only screening and searching for the terpene most active in relation to melanoma cells. The authors are considering undertaking further research on combinations of primarily the most active carvacrol with cytostatics.
Thank You very much for Your time for the Review and all critical comments. We hope that our responses to the objections are sufficient and satisfactory for the Reviewer.
Round 2
Reviewer 1 Report
Comments and Suggestions for Authors
The issues have been addressed